# Short-Term Tobacco Abstinence: Effects on Emotional Balance and Psychological Alienation

**DOI:** 10.3390/healthcare13141686

**Published:** 2025-07-14

**Authors:** Alean Al-Krenawi, Numan Al-Natsheh, Feras Ali Al-Habies, Ahmad Abudoush, Somaya Al-Ja’afreh, Ashraf Alqudah, Amal Salem Awawdeh, Dhaval Vinodkumar Patel

**Affiliations:** 1Resilience Research Centre, Dalhousie University, Halifax, NS B3H 4R2, Canada; 2Center for Research on Immigration and Settlement, Toronto Metropolitan University, Toronto, ON M5B 2K3, Canada; 3Department of Psychology, Al-Ahliyya Amman University, Amman 19328, Jordan; numan@ammanu.edu.jo; 4Department of Psychology, Faculty of Arts, The University of Jordan, Amman 11942, Jordan; firas4400@yahoo.com (F.A.A.-H.); a.abudoush@ju.edu.jo (A.A.); s.jaafreh@ju.edu.jo (S.A.-J.); a.alqudah@ju.edu.jo (A.A.); 5Center for Women’s Studies, The University of Jordan, Amman 11942, Jordan; a.awawdeh@ju.edu.jo; 6Department of Psychology, Algoma University, Sault Sainte Marie, ON P6A 2G4, Canada; dhapatel@algomau.ca

**Keywords:** tobacco abstinence, emotional balance, psychological alienation, smoking, tobacco, smoking cessation

## Abstract

Background/Objectives: Emerging evidence suggests that abstaining from tobacco smoking can influence emotional control and psychological well-being. This study examines the impact of short-term tobacco abstinence on emotional balance and psychological alienation through an experimental design. Methods: A total of 197 participants from a university in Jordan (academic year 2023/2024) were divided into three groups: one group abstained from smoking for 24 h (*n* = 65) and another for 48 h (*n* = 61), while the control group (*n* = 71) continued smoking as usual. Emotional balance and psychological alienation were assessed across all groups. Results: Participants who abstained from smoking (both 24 h and 48 h groups) reported lower scores on emotional balance and higher psychological alienation compared to the control group. Moreover, those in the 48 h abstinence group experienced significantly greater emotional imbalance and psychological alienation than those in the 24 h group. A significant negative correlation was found between emotional balance and psychological alienation in the 24 h abstinence and control groups, but not in the 48 h group. Conclusions: The findings indicate that short-term tobacco abstinence negatively affects emotional stability and increases feelings of psychological alienation. These effects are more pronounced after 48 h of abstinence compared to 24 h.

## 1. Introduction

Tobacco use remains a significant challenge for global public health policy. According to the World Health Organization (WHO), smoking is one of the leading causes of disease and death worldwide. It is related to a range of serious health issues, including cancer, cardiovascular diseases, and respiratory disorders. Globally, smoking accounts for nearly 7 million deaths annually and places a significant economic burden on healthcare systems due to direct treatment costs and productivity losses [1]. Epidemiological data suggest that tobacco use is disproportionately prevalent among individuals with lower socioeconomic status, lower educational attainment, and co-occurring mental health disorders [2,3,4]. Furthermore, the prevalence of smoking is notably higher among those with psychiatric conditions, particularly depression and anxiety [5,6,7]. Smokers with such comorbidities are more likely to experience difficulty quitting. These associations point to the importance of examining the psychological aspects of smoking behavior and withdrawal, which remain underexplored relative to the physical health outcomes.

Tobacco use remains a leading global health concern, not only for its long-term physical consequences but also for its effects on mental health and emotional functioning [1]. While nicotine is known to modulate mood and stress, its withdrawal—even over short periods—can cause irritability, anxiety, and emotional instability [7,8]. Short-term tobacco abstinence is associated with a range of acute psychological effects, which typically peak within the first 24 to 72 h. Research indicates that these effects can significantly impair mood and cognitive function, leading to irritability, anxiety, and restlessness [7,9]. Initially, individuals may feel more anxious and irritable within the first few days of quitting, often accompanied by increased tension and cravings for cigarettes; these symptoms usually last for around two weeks [10,11]. Although tobacco is often used to manage emotional discomfort, the psychological effects of abstinence do not necessarily mirror the relief associated with smoking. Rather than a direct reversal, nicotine withdrawal induces its own set of affective disturbances, which can disrupt emotional balance and increase relapse risk [12]. It is also important to distinguish between the emotional relief reported by smokers and the discomfort they experience upon abstaining. These are not conceptually symmetrical experiences and should be addressed as distinct phenomena in research and theory.

While many smokers perceive tobacco as a tool for managing emotional distress [13,14], this belief can obscure the paradoxical role that nicotine plays in mood dysregulation. The immediate “comfort” derived from smoking may not equate to proportional “discomfort” upon abstaining; instead, the neuroadaptation induced by chronic nicotine exposure can sensitize the individual to emotional instability during withdrawal [15]. 

Moreover, nicotine affects multiple neurotransmitter systems involved in emotion regulation, including dopaminergic and serotonergic pathways [16], thus further linking withdrawal to mood dysregulation.

In understanding the psychological consequences of tobacco use and cessation, it is essential to distinguish between three related constructs: emotional balance, emotional stability, and emotional regulation. Emotional balance refers to the overall harmony and equilibrium of a person’s emotional experiences, reflecting their ability to maintain steady moods without being overwhelmed by extremes [17,18]. It represents the outcome of both dispositional traits and dynamic regulatory processes.

Emotional stability, often considered a core personality trait, denotes the tendency to remain calm, composed, and less reactive to stress or negative affect [18,19,20]. High emotional stability is associated with reduced vulnerability to psychological distress, while low stability—commonly linked to neuroticism—has been associated with increased sensitivity to stress and mood fluctuations. In both clinical and occupational settings, emotional stability functions as a protective factor, enhancing resilience, performance under pressure, and resistance to impulsivity and substance use [18,19,21].

In contrast, emotional regulation encompasses the active processes and strategies individuals use to manage and modulate their emotional responses, such as controlling the intensity and expression of emotions [12,22]. While emotional stability provides a baseline temperament, emotional regulation involves dynamic skills to cope with emotional challenges, both of which influence one’s emotional balance over time. Understanding these distinctions is essential in the context of smoking, as tobacco use and cessation can disrupt emotional regulation and balance, contributing to withdrawal-related distress and psychological alienation [21,23]. According to the Gross model of emotion regulation [22], emotional regulation refers to the dynamic, conscious, and unconscious processes by which individuals influence their emotional experiences and expressions. In contrast, emotional balance is the resultant state reflecting an individual’s capacity to maintain emotional composure and psychological equilibrium. While successful regulation can lead to emotional balance, the two constructs are not interchangeable [17].

This distinction is particularly relevant in the context of smoking and nicotine withdrawal. Tobacco use alters neurochemical systems involved in mood regulation—particularly the dopaminergic and serotonergic pathways—contributing to the smoker’s emotional equilibrium [2,24]. Upon cessation, the disruption of these systems often leads to irritability, sadness, and affective instability, challenging both regulatory capacity and emotional balance. Such dysregulation contributes to the emergence of psychological alienation and withdrawal-related distress [21,23]. Understanding how emotional traits and skills respond to abstinence is thus vital for designing effective interventions and supporting long-term cessation success.

On the other hand, psychological alienation refers to a state of being disconnected from oneself, others, or society, typically characterized by isolation or detachment [23]. According to Seeman’s model, alienation encompasses five dimensions: powerlessness, normlessness, social isolation, self-estrangement, and meaninglessness. This model provides a robust framework for understanding alienation in smokers, particularly those experiencing internal conflict over smoking behavior [23,25].

In the context of smoking, psychological alienation is a common experience for smokers as well. Many smokers experience isolation, not just because of the social stigma linked to smoking, but also due to the inner struggle between wanting to quit and being dependent on nicotine [24]. This inner conflict can increase feelings of isolation, as smokers might feel separated from their peers and society in the form of social alienation [21]. Furthermore, the combination of physical and psychological impacts from smoking can result in a decreased feeling of self-value, adding to feelings of isolation. Additionally, individuals who smoke are at a higher risk of experiencing mental health conditions such as depression and anxiety, which are strongly associated with both emotional dysregulation and feelings of isolation [23]. While acute tobacco abstinence is associated with a broad spectrum of symptoms, including cognitive impairment, attentional bias, hunger, and heart rate fluctuations [7], the present study focuses specifically on emotional balance and psychological alienation. This focus is justified by emerging evidence that affective instability and perceived social disconnection may be among the most critical psychological predictors of relapse vulnerability.

Understanding the psychological effects of tobacco abstinence requires a grounding in current theories of emotional regulation. Gross’s process model of emotion regulation posits that individuals manage their emotional experiences through strategies such as situation selection, attentional deployment, cognitive reappraisal, and response modulation, all of which may be compromised during withdrawal when habitual coping (e.g., smoking) is removed [22]. Similarly, Koole’s integrative framework emphasizes the interplay between need-oriented, goal-oriented, and person-oriented regulation systems; abstinence may disrupt these systems and diminish emotional balance, especially in individuals with limited regulatory resources [26]. From a social-cognitive angle, self-affirmation theory suggests that when abstinence undermines a smoker’s self-image or sense of control, it may generate emotional distress or defensive reactions that erode emotional stability [27].

In parallel, the experience of psychological alienation during abstinence can be explained through Seeman’s classic model, which identifies alienation as a multidimensional state involving powerlessness, normlessness, and social isolation [28,29]. Feelings of alienation may worsen when individuals lose access to the social or psychological roles reinforced by smoking. Moreover, attachment theory helps to understand the variability in withdrawal responses: individuals with insecure attachment patterns are more likely to experience emotional dysregulation and interpersonal alienation in stressful situations, such as smoking cessation [30]. Together, these models clarify the empirical patterns observed in this study, namely, the deterioration of emotional balance and the increase in psychological alienation following short-term tobacco abstinence.

Prior research has mainly focused on the physical symptoms of nicotine withdrawal, including faster heart rate, increased appetite, and headaches. However, the psychological effects of nicotine withdrawal have received less attention. Nicotine withdrawal’s psychological consequences extend beyond physical symptoms, significantly impacting emotional balance and alienation [7,12,31]. Given the role that affective instability and perceived disconnection play in both dependence and relapse, there is a pressing need to explore how brief periods of abstinence affect these psychological dimensions. Understanding these effects could shed light on whether these psychological factors contribute to the mechanisms of the vicious cycle that compels smokers to continue smoking.

Study objectives:

The purpose of this study is to experimentally examine the psychological effects of short-term tobacco deprivation, with a particular focus on emotional balance and psychological alienation. The study aims to investigate whether brief smoking abstinence may exacerbate symptoms of emotional dysregulation and alienation by comparing smoker groups. This investigation addresses a significant gap in the current literature, which tends to emphasize physiological withdrawal symptoms over psychological responses. By centering emotional and interpersonal outcomes, the study offers new insights into the early stages of nicotine withdrawal that may be crucial for relapse prevention. Furthermore, understanding these effects can guide the development of tailored cessation programs that are sensitive to the emotional needs of smokers, particularly those with heightened vulnerability to affective instability or social disconnection.

Study hypothesis:

Based on prior research and theoretical frameworks, the current study hypothesizes that short-term tobacco abstinence will negatively affect emotional and psychological functioning. Specifically, we propose two hypotheses related to emotional balance and psychological alienation across varying durations of abstinence.

Participants who abstain from tobacco smoking for 24 or 48 h will exhibit significantly lower emotional balance and higher psychological alienation compared to participants in the control group, who continue smoking as usual.Participants in the 48 h abstinence group will report greater emotional imbalance and psychological alienation than those in the 24 h abstinence group.

## 2. Materials and Methods

### 2.1. Participants

The sample was recruited from a university in Jordan through on-campus advertisement. Recruitment was conducted through multiple channels on campus, including high-traffic areas (libraries, cafeterias) and digital platforms (university email lists and social media), ensuring broader reach beyond specific classes or groups. Participants were assured of confidentiality and anonymity to encourage honest and willing participation.

Recruitment resulted in a total of 237 students who smoked tobacco regularly and did not have a diagnosed mental disorder, which was an inclusion criterion. A total of 13 participants dropped out of the 24 h group, and 27 participants dropped out of the 48 h group. The final sample consisted of 197 participants, with ages ranging from 19 to 23 years, and 27.4% (*n* = 54) were female among them. They were randomly allocated into three groups: 24 h (*n* = 65), 48 h (*n* = 61), and a control group (*n* = 71).

Age and gender were included as covariates due to their well-documented influence on emotional regulation and psychological responses to tobacco use and withdrawal. Research has shown that younger individuals tend to exhibit greater emotional reactivity and are more susceptible to mood instability during nicotine abstinence [32,33]. Similarly, gender differences are prominent in smoking behavior and cessation outcomes; for example, women often report stronger emotional withdrawal symptoms and are more vulnerable to stress-induced relapse [34,35]. Controlling for these variables allows for a more accurate assessment of the direct impact of abstinence on emotional balance and psychological alienation, minimizing potential confounding effects.

### 2.2. Research Instruments

To measure emotional balance, the researchers used the Emotional Stability Scale developed by Ashour and Dukhan [36]. The scale consists of 23 statements and includes three dimensions: emotional control, emotional confrontation, and emotional flexibility. The scale is a five-point Likert scale: “Always applies, often applies, sometimes applies, rarely applies, and does not apply.” The scale has demonstrated good psychometric properties. Content validity was assessed through expert judgment, and construct validity was established by correlating each item with its respective dimension, yielding correlation coefficients ranging from 0.78 to 0.91. Reliability was assessed using Cronbach’s alpha, which yielded a value of 0.674 for the overall scale. Additionally, the study provided Cronbach’s alpha coefficients for individual dimensions, with 0.67 for emotional control, 0.88 for emotional confrontation, and 0.69 for emotional flexibility, indicating acceptable internal consistency and confirming the scale’s suitability for assessing emotional balance.

To measure psychological alienation, the study employed the Psychological Alienation Scale, developed by Tawafrah and Bani Yunus [24]. The scale is composed of 64 items that represent seven dimensions of psychological alienation: loss of a sense of belonging (items 1 to 9), noncompliance with standards (items 10 to 18), helplessness (items 19 to 28), lack of sense of value (items 29 to 36), loss of purpose (items 37 to 46), loss of meaning (items 47 to 53), and egocentrism (items 54 to 64). The scale is a five-point Likert scale: “Always applies, often applies, sometimes applies, rarely applies, and does not apply.” The reliability of the scale, as measured by Cronbach’s alpha coefficient, is 0.897.

### 2.3. Procedure

Following the recruitment process, participants were invited to an introductory session, where they had the opportunity to review the participant information sheet, gain a deeper understanding of the study purpose and procedures, and ask any questions they may have had. The participants then provided informed consent by signing the consent form before participation. After randomizing participants into three groups (i.e., 24 h abstaining, 48 h abstaining, control), the experiment began by giving participants instructions on whether to abstain from tobacco smoking for 24 h or 48 h, or not at all, with instructions to come to the researcher at the scheduled time to fill out the study-related tools (i.e., the control group and 24 h group were invited the next day, and the 48 h group were invited after 48 h).

### 2.4. Statistical Methods

All statistical analyses were conducted using RStudio software (2024.04.2 Build 764) [37]. Data were first screened for normality using the Shapiro–Wilk test and visual inspection of Q–Q plots. Given the non-normal distribution of key variables, non-parametric methods were employed throughout the analyses.

To compare emotional balance and psychological alienation across the three experimental groups (control, 24 h abstinence, and 48 h abstinence), the Kruskal–Wallis H test was used. When significant group differences were detected, post hoc pairwise comparisons were performed using the Mann–Whitney U test with Bonferroni correction to control for Type I error.

Subscale analyses were conducted for each construct to examine specific dimensions of emotional balance (emotional control, emotional confrontation, emotional flexibility) and psychological alienation (loss of belonging, noncompliance with norms, lack of purpose, egocentrism, and helplessness). The same non-parametric procedures were applied to these subscales.

Spearman’s rank-order correlation coefficients were calculated to assess the associations between emotional balance and psychological alienation within each group. To examine the potential influence of demographic covariates, a rank-based analysis of covariance (RANCOVA) was conducted with age and gender as covariates, using aligned rank transformation procedures suitable for non-parametric data.

Effect sizes for non-parametric comparisons were computed using the formula r = z/√*n*, to estimate the magnitude of observed group differences.

## 3. Results

The impact of short-term tobacco abstinence on emotional balance and psychological alienation was assessed using validated instruments across three experimental groups: a control group (*n* = 71), a 24 h abstinence group (*n* = 65), and a 48 h abstinence group (*n* = 61). Descriptive statistics revealed progressive changes in both constructs as the duration of abstinence increased.

Data were tested for normality using the Shapiro–Wilk test and Q–Q plots, which confirmed a non-normal distribution (*p* < 0.05), justifying the use of non-parametric procedures. Group differences in emotional balance and psychological alienation were examined using Kruskal–Wallis tests, which revealed highly significant differences across the three groups (χ^2^ = 168.93, df = 2, *p* < 0.001; χ^2^ = 163.03, df = 2, *p* < 0.001, respectively). These findings indicate that short-term tobacco abstinence significantly affects both constructs, with the most pronounced effects observed after 48 h of abstinence.

Post hoc comparisons with Bonferroni correction indicated statistically significant differences between all group pairs. Subscale analyses of emotional balance and psychological alienation confirmed these patterns, with the lowest emotional scores and highest alienation scores consistently found in the 48 h abstinence group. Helplessness was the only subscale that did not differ significantly between the 24 h group and the control group.

Spearman correlation analysis revealed a strong positive association between emotional balance and psychological alienation in the 24 h group (r = 0.908, *p* < 0.001), and negative correlations in both the control (r = –0.378, *p* = 0.001) and 48 h groups (r = –0.302, *p* = 0.018).

A rank-based ANCOVA (RANCOV) was conducted to examine the impact of age and gender on emotional balance. Age did not show a statistically significant effect, indicating no meaningful relationship between age and emotional stability within this sample. Gender, however, had a significant impact, with female participants scoring lower on emotional balance than male participants (β = –2.0, *p* = 0.033). The model accounted for a large proportion of variance in emotional balance scores (R^2^ = 88.3%), highlighting the importance of considering gender as a relevant factor in future studies on tobacco abstinence and emotional regulation.

Effect sizes for the primary comparisons were large (r = 0.62 for emotional balance; r = 0.58 for psychological alienation), supporting the robustness of the findings. Complete reporting of test statistics, degrees of freedom, and significance levels is presented in Table 1, Table 2 and Table 3.

The graphical data representation illustrated in Figure 1 shows clear trend lines for both within-group and between-group changes in emotional balance and psychological alienation over time. The visual representation supports the statistical findings, demonstrating a continuous decline in emotional balance and an increase in psychological alienation among abstinent individuals. These trends underscore the dynamic nature of psychological responses to tobacco abstinence and highlight the need for targeted support during the first 48 h post-smoking cessation.

## 4. Discussion

This study investigated the psychological consequences of short-term tobacco abstinence, focusing on emotional balance and psychological alienation—two constructs that have received limited attention in smoking cessation research. Our findings demonstrate that even brief periods of nicotine deprivation significantly impact emotional regulation and social connectedness, with effects intensifying over time. The observed differences across all experimental groups provide compelling evidence that withdrawal symptoms play a pivotal role in early psychological distress, which may serve as a precursor to dependence mechanisms.

The significant differences in emotional balance and psychological alienation scores between the control group and both abstinence groups support the hypothesis that nicotine withdrawal disrupts psychological homeostasis. Notably, the 48 h abstinence group exhibited the most pronounced decline in emotional stability and the highest levels of alienation, suggesting a progressive deterioration in psychological well-being during early abstinence. This aligns with previous research indicating that the initial 48–72 h following cessation represent a peak period for psychological withdrawal symptoms [7,38]. Our results corroborate findings from Welsch et al., who developed the Wisconsin Smoking Withdrawal Scale [39], and showed that psychological distress peaks within the first few days after quitting. However, our study adds depth by highlighting specific dimensions—particularly emotional flexibility, loss of purpose, and egocentrism—that are especially vulnerable during this phase. These findings reinforce the idea that emotional dysregulation and disconnection from self or others are not merely secondary consequences of nicotine withdrawal but core features of early cessation experiences.

One notable exception was the dimension of helplessness, where no statistically significant difference was found between the control group and the 24 h abstinence group. While differences were evident, they were less pronounced, possibly due to the relatively brief period of abstinence. This suggests that specific withdrawal symptoms, particularly those related to motivation and agency, may take longer than 24 h to manifest fully. It also underscores the non-uniform trajectory of nicotine withdrawal, consistent with the literature describing it as a dynamic and multifaceted syndrome [7,40].

A strong positive correlation between emotional balance and psychological alienation was observed in the 24 h abstinence group, a finding that initially appears paradoxical. However, this may reflect compensatory psychological processes. Some individuals may experience a heightened awareness of their emotional state upon initiating cessation, which temporarily increases perceived emotional regulation while simultaneously amplifying feelings of detachment from habitual coping strategies. In contrast, this relationship became weak and negative in the 48 h abstinence group, indicating a breakdown in emotional regulation and greater emotional volatility. This shift highlights the nonlinear nature of early withdrawal and supports the need for interventions that address both emotional and relational challenges at different stages of abstinence.

Neurobiologically, these changes are consistent with the known effects of nicotine on dopaminergic and serotonergic pathways [24,41]. Nicotine withdrawal has been shown to reduce prefrontal cortex activity and increase amygdala reactivity, contributing to irritability, anxiety, and impaired emotional control [42]. Our findings suggest that these neuroadaptations manifest behaviorally through increased alienation and reduced emotional resilience, particularly beyond the first 24 h of abstinence. Gender-specific responses were evident, with female participants reporting lower emotional balance scores compared to males. This aligns with recent studies suggesting that women may be more susceptible to mood-related withdrawal effects, potentially due to hormonal fluctuations, higher rates of comorbid depression, or differential coping strategies [39]. However, emerging evidence also indicates that women may derive greater long-term mental health benefits from smoking cessation than men [43], suggesting that although initial withdrawal may be more challenging for women, sustained abstinence could yield disproportionately positive outcomes. Individual variability in response to nicotine withdrawal further complicates the picture. Personality traits such as neuroticism, conscientiousness, and openness to experience have been identified as potential moderators of emotional and behavioral responses to abstinence [44]. Our data support the view that pre-existing psychological characteristics—rather than nicotine withdrawal alone—mediate the emotional impact of smoking cessation. This implies that cessation programs should incorporate personality assessments and tailor interventions accordingly to enhance effectiveness.

The present study was conducted among university students in Jordan, a demographic with unique sociocultural norms around smoking and emotional expression. Compared to national prevalence rates (~8.5% among women in Jordan), our sample included a higher proportion of female smokers (27.4%), which may reflect recruitment bias or shifting smoking patterns in urban youth populations. Future cross-cultural studies are needed to determine whether our findings can be generalized beyond similar settings. Additionally, our reliance on self-report measures introduces potential measurement bias, particularly in subjective domains like emotional balance and alienation. Although validated scales were used, future research would benefit from incorporating objective biomarkers (e.g., heart rate variability, cortisol levels) and behavioral indicators to triangulate findings and improve internal validity.

While much of the existing literature emphasizes physiological aspects of smoking cessation—such as appetite changes, weight gain, and metabolic shifts [45]—our focus on emotional and psychological dimensions fills a critical gap. Emotional imbalance and alienation are often overlooked in clinical models of withdrawal, despite their strong influence on relapse risk and treatment adherence. Furthermore, we observed a unique pattern in the 24 h abstinence group: emotional balance appeared slightly improved, while psychological alienation continued to rise. This discrepancy may reflect individual variability in coping mechanisms. For some individuals, the act of quitting may create a temporary sense of accomplishment or relief, masking underlying distress. However, as abstinence extends into the second day, emotional regulation deteriorates, and feelings of detachment become more salient. These findings contrast with some prior reports suggesting that emotional disturbances diminish more rapidly [41]. Such discrepancies may stem from methodological differences, including variations in sample composition, cultural context, or measurement tools. For instance, international studies often rely on mixed-gender or male-dominant samples, which may obscure gender-specific withdrawal profiles. Additionally, differing definitions of emotional balance—sometimes conflated with emotional regulation—may lead to inconsistent interpretations [12].

Study limitations:Despite its contributions, this study has several limitations that must be acknowledged:

Emotional balance and psychological alienation were assessed using self-reported questionnaires, which introduces the possibility of social desirability bias or subjective interpretation, potentially affecting the accuracy of the reported experiences.The 48 h abstinence period captures only the acute phase of withdrawal and does not allow for conclusions about longer-term psychological adaptation or the progression of psychological dependence over time.Smoking status was not verified through biochemical means such as cotinine testing, which could have enhanced the validity of the data and reduced the risk of misclassification.The sample composition included a disproportionately high percentage of female participants—27.4%—compared to the national prevalence of approximately 8.5% among women in Jordan, which may limit the generalizability of findings to broader populations, particularly in regions where smoking remains predominantly male.Furthermore, the majority of participants were university students, which restricts the applicability of results to other age groups, socioeconomic backgrounds, or cultural contexts.

Nonetheless, the study possesses notable strengths that support its scientific value. The experimental design allowed for causal inference by implementing a controlled abstinence protocol, distinguishing it from observational cessation studies and enhancing the ability to draw direct conclusions about the effects of tobacco deprivation. Moreover, validated and psychometrically sound scales were used to assess both emotional balance and psychological alienation, contributing to the reliability of the findings. Notably, the focus on early psychological changes during the initial 48 h of abstinence addresses a critical yet understudied phase in the quitting process, offering valuable insights into the dynamics of early withdrawal symptoms that are often overlooked in cessation research.

The findings underscore the significance of integrating psychological support into early smoking cessation initiatives, particularly within the first 48 h following tobacco use. During this critical period, emotional instability and feelings of detachment are likely to intensify, suggesting that timely interventions can significantly enhance the success rates of quitting. Healthcare providers are urged to acknowledge and address the emotional and relational challenges that accompany nicotine withdrawal by offering tailored strategies that align with individuals’ varying capacities for emotional regulation. Additionally, social support systems should be woven into the process to mitigate feelings of alienation and foster a sense of connectedness during the abstinence phase. Educating individuals about the anticipated psychological effects of withdrawal can empower them to more effectively prepare for and navigate common emotional obstacles. Furthermore, early monitoring of withdrawal symptoms is crucial to identify those at greater risk of relapse due to emotional distress, facilitating personalized and prompt interventions.

## 5. Conclusions

This study demonstrates that short-term tobacco abstinence significantly disrupts emotional balance and increases psychological alienation, particularly within the first 48 h. The findings suggest that psychological dependence on nicotine begins rapidly after cessation and escalates over time. Emotional and psychological disturbances, including mood instability and feelings of detachment, are prominent early withdrawal symptoms that can hinder cessation efforts. These effects are not uniform and may vary based on individual differences such as coping ability and personality traits. While early signs of psychological dependence may manifest within the 48 h window, we acknowledge that complete dependence is a more complex, long-term process, and our findings reflect only initial patterns of emotional change during early abstinence.

### Implications for Practice

The findings emphasize the critical need to address emotional and psychological symptoms in the early stages of smoking cessation. Health professionals and smoking cessation programs should take the following steps:Integrate psychological support into cessation interventions, particularly during the first 48 h, when emotional instability and feelings of isolation are at their peak;Tailor interventions to reflect individual differences in emotional regulation and coping capacities;Provide education on the anticipated psychological effects of withdrawal to prepare individuals for common emotional challenges;Incorporate social support strategies to mitigate feelings of alienation and enhance social connectedness during the abstinence period;Closely monitor early withdrawal symptoms to identify individuals at risk of relapse due to emotional distress, allowing for timely and personalized support.

By adopting these practices, clinicians and public health professionals can improve the effectiveness of smoking cessation efforts and enhance long-term outcomes for those trying to quit.

## Figures and Tables

**Figure 1 healthcare-13-01686-f001:**
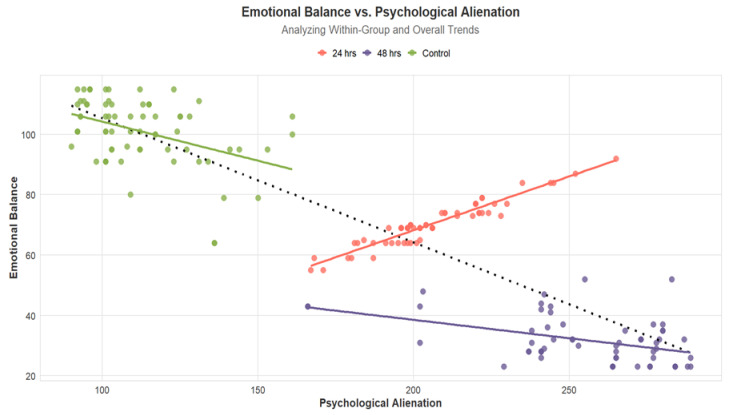
Plotting of within-group and between-group trend lines.

**Table 1 healthcare-13-01686-t001:** Kruskal–Wallis and pairwise comparisons of emotional balance with adjusted *p*-values using Bonferroni correction.

Emotional Balance Dimensions	Kruskal–Wallis	Pairwise Comparison (24–48)	Pairwise Comparison (24–Control)	Pairwise Comparison (48–Control)
Overall emotional balance	χ^2^ = 168.93, df = 2, *p* < 0.001 ***	Z = 6.41 *p*. adj < 0.001 ***	Z = −6.55 *p*. adj < 0.001 ***	Z = −12.99 *p*. adj < 0.001 ***
Emotional control	χ^2^ = 169.41, df = 2, *p* < 0.001 ***	Z = 6.41 *p*. adj < 0.001 ***	Z = −6.57 *p*. adj < 0.001 ***	Z = −13.01 *p*. adj < 0.001 ***
Emotional confrontation	χ^2^ = 169.32, df = 2, *p* < 0.001 ***	Z = 6.41 *p*. adj < 0.001 ***	Z = −6.57 *p*. adj < 0.001 ***	Z = −13.00 *p*. adj < 0.001 ***
Emotional flexibility	χ^2^ = 168.46, df = 2, *p* < 0.001 ***	Z = 6.42 *p*. adj < 0.001 ***	Z = −6.52 *p*. adj < 0.001 ***	Z = −12.97 *p*. adj < 0.001 ***

*** significant at *p* < 0.001. S: significant; NS: non-significant.

**Table 2 healthcare-13-01686-t002:** Kruskal–Wallis and pairwise comparisons of psychological alienation with adjusted *p*-values using Bonferroni correction.

Psychological Alienation Dimensions	Kruskal–Wallis	Pairwise Comparison (24–48)	Pairwise Comparison (24–Control)	Pairwise Comparison (48–Control)
Overall psychological alienation	χ^2^ = 163.03, df = 2, *p* < 0.001 ***	Z = −5.24 *p*. adj < 0.001 ***	Z = 7.43 *p*. adj < 0.001 ***	Z = 12.66 *p*. adj < 0.001 ***
Loss of a sense of belonging	χ^2^ = 147.97, df = 2, *p* < 0.001 ***	Z = −3.49 *p*. adj < 0.001 ***	Z = 8.32 *p*. adj < 0.001 ***	Z = 11.74 *p*. adj < 0.001 ***
Noncompliance with standards	χ^2^ = 171.42, df = 2, *p* < 0.001 ***	Z = −5.95 *p*. adj < 0.001 ***	Z = 7.09 *p*. adj < 0.001 ***	Z = 13.05 *p*. adj < 0.001 ***
Helplessness	χ^2^ = 116.63, df = 2, *p* < 0.001 ***	Z = −10.41 *p*. adj < 0.001 ***	Z = −2.74 *p*. adj = 0.018 *	Z = 7.93 *p*. adj < 0.001 ***
Lack of a sense of value	χ^2^ = 158.78, df = 2, *p* < 0.001 ***	Z = 4.94 *p*. adj < 0.001 ***	Z = 12.50 *p*. adj < 0.001 ***	Z = 7.25 *p*. adj < 0.001 ***
Loss of purpose	χ^2^ = 154.74, df = 2, *p* < 0.001 ***	Z = −4.33 *p*. adj < 0.001 ***	Z = 7.91 *p*. adj < 0.001 ***	Z = 12.19 *p*. adj < 0.001 ***
Loss of meaning	χ^2^ = 158.9, df = 2, *p* < 0.001 ***	Z = −5.45 *p*. adj < 0.001 ***	Z = 7.08 *p*. adj < 0.001 ***	Z = 12.53 *p*. adj < 0.001 ***
Egocentrism	χ^2^ = 165.17, df = 2, *p* < 0.001 ***	Z = −5.40 *p*. adj < 0.001 ***	Z = 7.37 *p*. adj < 0.001 ***	Z = 12.76 *p*. adj < 0.001 ***

* Significant at *p* < 0.05, *** significant at *p* < 0.001. NS: non-significant.

**Table 3 healthcare-13-01686-t003:** Correlation between psychological alienation and emotional balance using the Spearman correlation coefficient.

Variables	Abstaining Time	r-Value	*p*-Value
Psychological Alienation and Emotional Balance	24 h	0.908	<0.001 ***
	48 h	−302	0.018 *
	Control	−378	0.001 **

* Significant at *p* < 0.05, ** significant at *p* < 0.01, *** significant at *p* < 0.001

## Data Availability

The original data presented in the study are openly available on FigShare at https://doi.org/10.6084/m9.figshare.29161604.

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
