# Peer review of "Short-Term Tobacco Abstinence: Effects on Emotional Balance and Psychological Alienation"

_healthcare, 2025, doi:10.3390/healthcare13141686_

Round 1
Reviewer 1 Report
Comments and Suggestions for Authors
Dear Authors,
Thank you for the opportunity to review your manuscript. The topic is undoubtedly relevant and the study presents several valuable insights. However, I would like to suggest a number of important revisions to improve the clarity, coherence, and methodological transparency of your work.
To begin with, the Introduction would benefit from a clearer structure and stronger focus. As currently written, it occasionally feels disjointed, with general information about smoking, mental health effects, sociodemographic factors, and psychological constructs (emotional balance and alienation) presented in a fragmented manner, without sufficient conceptual integration. The transition to the focus on emotional balance and psychological alienation (lines 68–69) appears abrupt and lacks adequate theoretical grounding.
Recommendation: Structure the introduction according to the logic of: (1) broad context, (2) specific problem, (3) research gap (what has not yet been studied, what remains unclear), and (4) study objective. At the end of the introduction, consider adding a clear subheading explicitly stating the research aims and hypotheses.
In the Research Instruments section, the label "Instrument 1" should be removed and replaced with the exact name of the instrument. It is also important to specify whether the questionnaires used include subscales, how the items are scored (e.g., sum score, average), and what each subscale consists of.
Recommendation: Begin with the full and correct names of the instruments, indicate the number and nature of subscales, the number of items per subscale, and the exact method used for calculating total scores. Future researchers should be able to fully replicate the study based on this section.
Currently, statistical methods are not described in sufficient detail. A separate subsection on statistical analysis should be created, listing all statistical tests used and explaining their rationale. As the results suggest the use of non-parametric tests, it is essential to clarify why these tests were chosen.
Recommendation: Clearly separate statistical methods under their own subheading. Specify all tests used, the software applied, and the alpha threshold (e.g., p < 0.05). Justify the choice of non-parametric methods (e.g., due to sample distribution, ordinal data, etc.).
Regarding the Results, the manuscript lacks basic descriptive statistics for the questionnaires used. The current tables are difficult to interpret—especially for readers without advanced statistical knowledge.
Recommendation: Present tables in a cleaner and more standardized format, with clearly labeled rows and columns. Long textual labels should be shortened and made consistent (e.g., use “p.adj” rather than “p. Adj”). Consider including only adjusted p-values for clarity. Footnote symbols (*, **, ***) are mentioned but do not appear in the actual table—please ensure consistency. In order to facilitate comparisons between groups, it would be advisable to report medians and interquartile ranges for each group (or means and standard deviations, depending on data distribution). In Table 3, the final column may be unnecessary; if the significance threshold is stated in the methods section (e.g., p < 0.05), the reader can interpret significance accordingly. Alternatively, significant values may be bolded for emphasis.
Additionally, on line 168, the term "p-value of .033" should be replaced with standard notation (p = 0.033) for clarity and consistency.
Moving on to the Discussion, lines 195–218 mostly summarize results in relation to previous studies. However, there is a lack of critical reflection: Why are the results similar? Were the same instruments and methodologies used? How comparable are the participant samples? These are important questions that require a more in-depth analysis.
Also, lines 226–230 briefly mention that some studies found faster symptom resolution than what was observed here. This is a crucial observation but is underdeveloped. The discussion would benefit from exploring possible explanations: Are these differences due to methodology, sample characteristics, cultural context, or length of abstinence?
A further shortcoming is the complete absence of any reflection on the limitations of the study. Issues such as sample size, reliance on self-report instruments, the limited 48-hour abstinence window, potential habituation effects, and the gender composition of the sample are not addressed. This is a significant omission that needs to be corrected.
Lastly, the conclusions about the development of psychological dependence within just 48 hours (lines 185–186, 248–250) appear overstated given the temporal limitations of the study. These claims should be framed more cautiously, with a note that longitudinal research is required to draw firm conclusions about the onset and progression of dependence.
In summary, the manuscript shows promise, but revisions are necessary to strengthen its conceptual clarity, methodological transparency, and interpretive depth.
Reviewer 2 Report
Comments and Suggestions for Authors
Thank you for the opportunity to review this manuscript.
Introduction
- Please provide scientific evidence documenting the demographic differences between the general population and smokers (p. 2, lines 51-53).
- Some sentences contain logical gaps. For example, 'However, since psychological dependence is closely related to the comfort and relief smoking provides, it is crucial to further investigate the impact of abstaining from smoking on discomfort levels...' (p. 3, lines 95-97). The logical problem is that this assumes discomfort from abstaining is equivalent to the comfort that smoking provides. These are not necessarily equivalent concepts. The comfort someone derives from smoking does not automatically translate to proportional discomfort when they cease smoking. Furthermore, emotional balance levels do not directly reflect 'discomfort levels.'
- Previous research found that 'the acute tobacco abstinence syndrome is not a monotonic phenomenon, but effects are largest for craving, subjective attentional bias, negative affect, overall withdrawal severity, concentration difficulty, hunger, and heart rate.' This study focuses only on psychological domain phenomena, which may impede a multifaceted understanding of tobacco abstinence syndrome.
Materials and Methods
- According to a 2017 World Health Organization (WHO) survey, approximately 8.5% of Jordanian women were smokers. However, the proportion of female smokers in this study is 27.4%. This sample may therefore be over-representative of female smokers.
- On-campus recruitment may compromise survey reliability, including low response rates and difficulty reaching specific demographics. How were these problems addressed?
- What do the values '0.05 or 0.01' represent here? (p. 3, line 119)
- The citation 'Ashour & Dukhan, 2017' (p. 3, line 115) is missing from the references.
- Please provide complete demographic characteristics of the statistical sample and describe the statistical methods used.
Results
- Why were age and gender considered as covariates? What was the rationale for using rank-based ANCOVA for covariance analysis?
Discussion
- Some statements contradict the results. For example, 'A strong positive correlation between emotional balance and psychological alienation was observed...' (p. 7, lines 195-196). However, according to Figure 1, this correlation is negative in some groups.
Reviewer 3 Report
Comments and Suggestions for Authors
REVIEW
“Short-Term Tobacco Abstinence: Effects on Emotional Balance and Psychological Alienation”
The article examines a question of considerable scientific and clinical relevance: the short-term psychological effects of quitting smoking, focusing particularly on variables such as emotional balance and psychological alienation. This is a timely and relevant approach, given the persistent prevalence of smoking worldwide and its well-known psychosocial implications, especially among young people, where smoking is often intertwined with processes of identity, emotional regulation and social belonging.
However, from a critical perspective, the theoretical basis of the study is not very consistent. Although essential concepts such as psychological alienation and emotional balance are introduced, their approach is superficial and poorly articulated. There is a lack of in-depth exploration of the underlying mechanisms, as well as a critical and updated review of the available theoretical models. Many of the statements are supported by previous references, but these are mostly secondary in nature and are not anchored in solid conceptual frameworks that allow the phenomenon under discussion to be placed within a robust psychological architecture.
A descriptive empirical orientation predominates over analytical argumentation, which weakens the reader's ability to understand the logic behind the selection of certain variables. Similarly, although relevant correlations between constructs are identified, their theoretical significance is not explored in depth, nor are hypotheses about possible mediating or moderating variables considered, which significantly reduces the explanatory potential of the study.
As for the formal structure of the manuscript, it is correctly organised according to the IMRyD format (Introduction, Method, Results and Discussion), although there are some problems with the English wording, including the unnecessary or incorrect use of connectors such as ‘At first’, grammatical errors and the presence of redundant or poorly articulated sentences that compromise the clarity and fluency of the text in certain passages.
The introduction provides a general context for the problem of smoking, but devotes little space to specifically and argumentatively justifying the central hypothesis of the study. For its part, the discussion section, despite presenting the results, lacks a critical and nuanced interpretation of the findings: they are not contrasted in depth with the previous literature, nor is the directionality or magnitude of the observed effects clearly discussed, which limits its scope and theoretical projection.
In terms of academic style, although the use of psychological terminology is generally appropriate, significant conceptual inaccuracies are detected. An example of this is the equating of ‘emotional regulation’ with ‘emotional balance’, two constructs that in the specialised literature have fundamental distinctions in both their definition and operationalisation.
The experimental design between groups represents, in principle, a methodological strategy consistent with the objective of the study. The inclusion of a randomisation procedure and the attempt to control for confounding variables by excluding participants diagnosed with mental disorders are positively evaluated. However, the design suffers from several relevant limitations:
- Internal validity: It is not specified whether any biochemical verification of abstinence was carried out, such as measuring cotinine levels, which seriously compromises the fidelity of the experimental treatment.
- Sample: Although the sample size (n = 197) is reasonable, the representativeness of the study is limited, given that all participants are Jordanian university students, with a clearly unbalanced gender distribution (72.6% men), which introduces a potential bias in the results.
- Instruments: The scale used to assess emotional balance has low internal reliability (α = 0.674), which is barely acceptable according to psychometric standards. This weakness calls into question the accuracy of the measurements. In contrast, the alienation scale used shows high internal consistency (α = 0.897), which is a strength of the study.
- Statistical analysis: The choice of the Kruskal-Wallis test is appropriate given the non-normality of the data, and the application of post hoc comparisons with Bonferroni correction demonstrates basic technical knowledge. However, the inclusion of multivariate analyses to control for possible cross-effects is lacking. Furthermore, no data on effect sizes are provided, which makes it difficult to interpret the practical relevance of the findings.
With regard to the bibliographic base, a significant limitation should be noted: most of the sources cited are from before 2010, with few references from the last five years. This deficiency is striking considering the considerable volume of recent studies addressing the connections between smoking and mental health from contemporary perspectives.
Despite its limitations, the article provides interesting empirical data on a little-explored dimension: the immediate psychological effects of quitting smoking. This approach, still emerging in the scientific literature, opens the door to new lines of research. The experimental design is relevant and the analyses performed are technically correct, although they could be optimised. However, the research is hampered by a lack of theoretical depth, certain methodological weaknesses and writing that requires substantial improvement to meet the standards of clarity and rigour expected of an academic publication of impact.
The conclusions, while reasonable in their approach, would benefit from further elaboration, especially with regard to their possible clinical or preventive implications in the field of mental health.
Specific suggestions for improvement:
- Deepen the theoretical framework by including current models of emotional regulation and alienation, such as self-affirmation theories or attachment-based approaches to tobacco use.
- Replace the emotional balance scale with more psychometrically validated versions that have higher levels of internal consistency.
- Correct the editorial errors detected, paying particular attention to syntax and terminological clarity.
- Incorporate objective measures of abstinence verification, such as nicotine biomarkers, to increase the internal validity of the study.
- Update the bibliography used, ensuring strict compliance with the Vancouver style and the inclusion of relevant literature from the last five years.
- Add effect size analysis and explore multivariate models to enrich the interpretation of the results and strengthen the conclusions drawn.
Comments on the Quality of English Language
It would be advisable to strengthen your command of English, especially in its academic register, to ensure clear, accurate writing that is free of errors that could affect the quality of the text.
Round 2
Reviewer 1 Report
Comments and Suggestions for Authors
Dear Authors,
Thank you for the detailed explanations of the ambiguities and for the corrections made. However, the following additional changes should be implemented:
Statistical Methods This section needs to be separated from the "Results" and placed under its own subheading, "Statistical Methods," which should follow the "Methods" section. Currently, these details have become part of the results, and they should definitely be separated. This part, which describes the application of tests, normality of distribution, and software used, should be clearly delineated within this new subheading.
Furthermore, above Table 1, only the most important results should be described.
Reviewer 2 Report
Comments and Suggestions for Authors
The authors have addressed all of my concerns. The paper is ready for publication.
Author Response
Thank you for your response.
Reviewer 3 Report
Comments and Suggestions for Authors
He revisado el nuevo artículo que enviaste (“Short-Term Tobacco Abstinence: Effects on Emotional Balance and Psychological Alienation”) y lo comparo con las pautas metodológicas y de publicación revisadas.
Introducción
Bien contextualizada con cifras de la OMS y literatura reciente.
Diferencia claramente entre equilibrio emocional, estabilidad y regulación.
Algunas referencias clave son algo antiguas (por ejemplo, Seeman para alienación).
Método
Diseño experimental claro, con tres grupos y aleatorización.
Buena justificación de las covariables (edad y género).
Alta tasa de abandono en los grupos de abstinencia (13 y 27 sujetos), sin análisis de impacto (p.ej., análisis de intención de tratar).
El alfa de la escala de estabilidad emocional es bajo (0.674), cuestionando su consistencia interna.
Resultados
En el fragmento proporcionado no aparecen tablas ni detalles estadísticos. Para futuras ediciones seria importante indicar los tamaños del efecto, intervalos de confianza y exactitud de los p-valores para apoyar las conclusiones.La interpretación parece lineal (abstinencia causa malestar), pero no se descarta adecuadamente la posibilidad de factores de confusión
El artículo sí ha mejorado notablemente en rigor teórico, claridad de hipótesis y diseño experimental respecto a una versión previa hipotética. No obstante, podría aún tener mejoras en aspectos metodológicos, estadísticos y redacción.
Author Response
Thank you for your response.